# Enhanced Levels of Glycosphingolipid GM3 Delay the Progression of Diabetic Nephropathy

**DOI:** 10.3390/ijms241411355

**Published:** 2023-07-12

**Authors:** Shokichi Naito, Kenichi Nakayama, Nagako Kawashima

**Affiliations:** 1Department of Nephrology, Kitasato University School of Medicine, 1-15-1 Kitasato, Minami, Sagamihara 252-0374, Kanagawa, Japan; snaito@med.kitasato-u.ac.jp; 2National Institute of Advanced Industrial Science and Technology (AIST), 1-1-1 Umezono, Tsukuba 305-8560, Ibaraki, Japan; k-nakayama@aist.go.jp

**Keywords:** glycosphingolipid GM3, valproic acid, diabetic nephropathy, albuminuria, podocyto injury

## Abstract

We recently found that albuminuria levels in patients with minimal change disease (MCD) and focal segmental glomerulosclerosis (FSGS) inversely correlate with glycosphingolipid GM3 expression levels in glomerular podocytes. Moreover, we showed enhanced expression of GM3 via activation of the GM3 synthase gene upon administration of valproic acid (VPA) is effective in suppressing albuminuria and podocyte injury in mice with anti-nephrin antibody-induced podocytopathy. However, the therapeutic effect of GM3 on diabetic nephropathy, which is the most common underlying disease in patients undergoing dialysis and with podocyte injury, remains unclear. Here, we investigated the therapeutic effect of enhanced GM3 expression via VPA on podocyte injury using streptozotocin-induced diabetic nephropathy model mice. Administration of VPA clearly decreased levels of albuminuria and glomerular lesions and inhibited the loss of podocytes and expansion in the mesangial area. Furthermore, we found that albuminuria levels in patients with diabetic nephropathy inversely correlate with the expression of GM3 in podocytes. These results indicate that maintaining GM3 expression in podocytes by administration of VPA may be effective in treating not only podocyte injury, such as MCD and FSGS, but also the late stage of diabetic nephropathy.

## 1. Introduction

Glycosphingolipids expressed on the cell membrane play a role in transducing and regulating various extracellular stimuli into intracellular activities. Glycosphingolipids are abundantly expressed throughout the body of vertebrates, especially in the brain and nervous system. Intriguingly, dynamic changes to glycosphingolipid structures occur during development and differentiation [1,2,3]. Furthermore, “cancer-related antigens” expressed in various cancer cells and tissues have been shown to comprise sugar chains of glycosphingolipids such as GD3, GM2, and GD2 [4,5,6] (Figure 1). Among them, GM3 (Neu5Acα2,3Galβ1,4Glcβ1,1Cer), which is made up of glycosphingolipids with sialic acid, regulates the phosphorylation of receptors such as EGFR as well as insulin receptors [7,8,9,10]. VPA is an up-regulator of the GM3 synthase gene (*ST3GLA5*, *St3gal5*; CMP-NeuAc: lactosylceramide α2,3-sialyltransferase; EC 2.4.99.9), which results in elevated levels of GM3 [10,11,12].

Recently, the characteristics and functions of glycosphingolipids in the kidney have been reported in several studies. Histochemical and immunoelectron microscopic analysis identified GM3 expressed on podocyte cell membranes in adult human renal tissue [13]. Another study reported that the receptor for soluble VEGF (sFlt-1) is GM3 in glomerular podocytes due to the binding of GM3 and sFlt-1 on podocytes and that reduced GM3 expression may result in proteinuria [14]. Because podocyte dysfunction is directly correlated with the progression of proteinuric chronic kidney disease (CKD) [15], GM3 may be a therapeutic target for this condition.

Our previous studies have mainly focused on how glycosphingolipids on the cell membrane regulate the functions of membrane proteins, such as the epidermal growth factor receptor [7,9,10,16]. Recently, we investigated the relationship between podocyte injury and GM3 and demonstrated that in healthy subjects, GM3 colocalizes with nephrin, which plays an important role in the glomerular filtration barrier, whereas there was an inverse correlation between albuminuria levels and not only nephrin but also GM3 expression levels in patients with minimal change disease (MCD) and focal segmental glomerulosclerosis (FSGS) due to podocyte injury [17]. Furthermore, using mice with anti-nephrin antibody-induced podocytopathy, we demonstrated that preventive administration of valproic acid (VPA) significantly inhibited albuminuria and podocyte injury. However, GM3 synthase-deficient mice were highly sensitive to the anti-nephrin antibodies, and administration of VPA failed to inhibit albuminuria and podocyte injury. In addition, in vitro analysis revealed that GM3 directly binds to nephrin, and GM3 present in the cell membrane of podocytes colocalizes with nephrin to act together. Based on these results, GM3 was shown to be essential for the maintenance of normal kidney functions via nephrin and podocytes [18].

Diabetic nephropathy is one of the three major complications of diabetes mellitus. This complication generally occurs in diabetic patients with insufficient long-term blood glucose control [19]. Furthermore, diabetic nephropathy is known to cause albuminuria and renal dysfunction, leading to decreased podocyte function [20]. However, little is known about the function of glycosphingolipids, especially gangliosides, in diabetic nephropathy, which remains unclear.

The purpose of this study was to examine the therapeutic potential of glycosphingolipid GM3 in diabetic nephropathy. Here, we analyzed the therapeutic effects of enhanced GM3 via VPA-induced activation of *St3gal5* on albuminuria and podocyte injury in streptozotocin (STZ)-induced diabetic nephropathy model mice. We also examined the level of nephrin and GM3 expression in the glomeruli and the correlation between the expression level of nephrin and GM3 and the degree of proteinuria in both healthy subjects and patients with diabetic nephropathy. Our findings suggest that maintaining GM3 expression via administration of VPA may be an effective treatment for the protection of podocytes not only for podocytopathies such as MCD and FSGS but also for late-stage diabetic nephropathy.

## 2. Results

### 2.1. Characterization of the STZ-Induced Diabetic Model in Mice

In this study, type 1 diabetic model mice were induced with 100 µL of 10 mg/mL STZ administered into the peritoneal cavity for five days (Figure 2A). Two weeks after the first STZ administration, mice with a blood glucose concentration of over 280 mg/dL were used as STZ-induced diabetic model mice for the subsequent animal study. Oral administration of VPA (4 mM drinking water) was started 1 week after day 5 of STZ administration and continued for approximately 4 months. Urine and serum were sampled on days 60 and 120 from the start of the study, and kidney tissues were collected on day 120 (Figure 2A). In terms of body weight, the diabetic nephropathy (DN) group showed a marked decrease, while the VPA-treated group after the onset of diabetic nephropathy (DN + VPA) showed a slightly slower rate of decrease (Figure 2B). No significant difference was observed between the untreated (control) group and the group given just VPA (VPA-only). For the DN group and the DN + VPA group, blood glucose levels continued to rise significantly from 20 days after STZ administration and showed a high level (>900 mg/dL) after 120 days (Figure 2C). Moreover, elevated blood glucose levels and a marked decrease in insulin secretion in pancreatic islet cells were similar in both the DN and DN + VPA groups (Figure 2D). Albuminuria on day 120 was 1.70 ± 1.46 g/g Cr (control 0.07 ± 0.07 g/g Cr, *p* < 0.05: control vs. DN) in the DN group compared to 0.43 ± 0.26 g/g Cr in the diabetic DN + VPA group (*p* < 0.05: vs. DN, *p* < 0.01: vs. untreated) (Figure 2E). The level of serum creatine at day 120 was also increased in the DN group (control 0.30 ± 0.04 mg/dL) compared to the untreated (control) group (control 0.14 ± 0.02 mg/dL, *p* < 0.01: control vs. DN), but decreased in the DN + VPA group (0.20 ± 0.05 mg/dL) (Figure 2F). Levels of albuminuria and serum creatinine in the VPA-only group were not significantly different from those in the control group (Figure 2E,F). Taken together, these results showed that, despite late-stage diabetic nephropathy being induced 120 days after STZ administration, the level of albuminuria due to diabetic nephropathy in the DN + VPA group was clearly suppressed compared to the DN group (Figure 2E).

### 2.2. Analysis of Glomerular Morphological Changes and Podocyte Injury

Next, potential morphological differences in kidney glomeruli from each group were investigated using various staining procedures (Figure 3 and Figure 4). Although the DN and the DN + VPA groups both showed glomerular hypertrophy and mesangial expansion (Figure 3B and Figure 4A,C,D), but it was clearly less marked in the latter case. There was no mesangial expansion in the control or VPA-only groups (Figure 3B and Figure 4D). These results showed that continuous administration of VPA after destruction of pancreatic islet cells can delay morphological changes in glomeruli, such as glomerular hypertrophy and mesangial expansion, which are otherwise observed in late-stage STZ-induced diabetic nephropathy. However, no significant fibrosis was evident in the renal interstitium of all the groups (Figure 3C). 

Next, the protective effect of podocytes was evaluated by measuring the number of podocytes by p57 staining (Figure 4A,B). In the DN group, there was a significant decrease in the number of podocytes compared to the control (DN 18.7 × 10^2^ ± 4.2 /µm^2^, untreated 35.1 × 10^2^ ± 4.1 /µm^2^: *p* < 0.01). However, by comparison to the DN group, the decrease in the number of podocytes was significantly less marked for the DN + VPA group (28.8 × 10^2^ ± 5.0/µm^2^) (*p* < 0.01) (Figure 4A,B). Furthermore, glomerular hypertrophy was evident in both the DN and DN + VPA groups (Figure 4A,C). However, whereas there was a significant expansion of mesangial cells in the DN group, this was less marked for the DN + VPA group (DN 58.2 ± 11.1%, DN + VPA 23.9 ± 9.4%: *p* < 0.01) (Figure 4A,D). These data suggest that VPA elicits a therapeutic effect on diabetic nephropathy by suppressing podocyte injury.

### 2.3. The Expression of Platelet-Derived Growth Factor Receptor-β (PDGFR-B) in Glomeruli

Activation of platelet-derived growth factor receptor-β (PDGFR-β) is associated with the progression of diabetic nephropathy [21]. Therefore, we investigated the effect of treatment with VPA on the expression level of PDGFR-β. Our results show that PDGFR-β was expressed at significantly elevated levels in the DN group (23.9 ± 8.5%: *p* < 0.01 vs. all) compared to all the other groups. Moreover, the expression levels of PDGFR-β for the DN + VPA group (12.7 ± 2.7%) were similar to those of the untreated (control) group (10.3 ± 4.1%, *p* = 0.20) (Figure 5A,B). It is known that PDGFR-β is specifically expressed in mesangial cells [22], which tend to undergo hypertrophy during diabetes mellitus. These results indicate that PDGFR-β was highly expressed in the significantly enlarged mesangial area of the glomeruli in the DN group but suppressed in the DN + VPA group (Figure 5A,B). Thus, these results also indicate that VPA has a therapeutic effect on diabetic nephropathy. 

### 2.4. Analysis of GM3 and Nephrin Expression in Kidney Glomeruli

Next, we examined the correlation between the reduction of glomerular lesions and symptoms in the DN + VPA group (Figure 2, Figure 3, Figure 4 and Figure 5). Specifically, we analyzed the expression levels of GM3 and nephrin in the glomeruli of each group. In the normal kidney, nephrin and GM3 colocalize within the glomerulus [18]. However, levels of both nephrin and GM3 in podocytes were dramatically decreased in the DN group by comparison to the untreated group (*p* < 0.01 for both groups) (Figure 6A–D). The level of nephrin expression in the DN + VPA group was the same as the control group, despite being under diabetes mellitus (Figure 6A). GM3 expression was increased in the VPA group compared to the control group (Figure 6A–D). Intriguingly, GM3 expression was enhanced with no negative effect on nephrin in the VPA-only group (Figure 6A–D).

### 2.5. Expression of Nephrin and GM3 and Their Correlation with Albuminuria in Patients with Diabetic Nephropathy 

We examined the expression levels of nephrin and GM3 in the glomeruli of healthy subjects and diabetic nephropathy patients and their correlation with proteinuria (Figure 7 and Table 1). The baseline characteristics showed that hemoglobin, serum albumin, and eGFR were significantly lower and serum urea nitrogen and serum creatinine were higher in diabetic nephropathy patients compared to healthy subjects (*p* < 0.01 for all). These data indicate that diabetic nephropathy is associated with nephrotic-range proteinuria and poor kidney function. Proteinuria was significantly higher in diabetic nephropathy patients (4.94 [1.36–8.79] g/g Cr) by comparison to healthy subjects (0.04 [0.01–0.06] g/g Cr, *p* < 0.01) (Table 1). In diabetic nephropathy, proteinuria is a strong prognostic factor for kidney function, as is albuminuria [23]. Furthermore, both nephrin and GM3 expression in diabetic nephropathy patients were markedly decreased compared to healthy subjects (*p* < 0.01 for both nephrin and GM3) (Figure 7B,C). Proteinuria, measured at the time of kidney biopsy, was found to be inversely correlated with nephrin expression (r = −0.7364). There was also a corresponding inverse correlation with GM3 expression (r = −0.4455). These results suggest that in diabetic nephropathy, as in MCD and FSGS, proteinuria increases with decreased GM3 expression.

## 3. Discussion

Diabetic nephropathy is one of the three major complications of diabetes mellitus resulting from prolonged hyperglycemia. The disease is asymptomatic in its early stages, but as kidney function declines, various symptoms appear, ultimately leading to end-stage kidney disease and the need for renal replacement therapy. Given that diabetic nephropathy is the leading cause of patients having to undergo dialysis, prevention of this disease is an urgent issue [24].

In this study, levels of the glycosphingolipid GM3 were found to be decreased in glomeruli during STZ-induced late-stage diabetic nephropathy in model mice. However, administration of VPA inhibited albuminuria and podocyte injury, which are otherwise associated with this condition, by enhancing GM3 expression in the podocyte cell membrane. Furthermore, we revealed that decreased GM3 expression as well as nephrin in podocytes inversely correlated with proteinuria in patients with diabetic nephropathy. Our previous study demonstrated the efficacy of GM3 for treating kidney disease, especially podocytopathy [18]. Here, we show that GM3 is also effective in treating diabetic nephropathy.

Prolonged hyperglycemia leads to the appearance of various cellular lesions in glomeruli and the development of diabetic nephropathy. Several lines of evidence suggest a correlation between diabetic nephropathy and glycosphingolipids. In 1993, Zador et al. analyzed glycosphingolipid species in the whole kidney of a rat model during the early stage of STZ-induced diabetic nephropathy [25]. Furthermore, Kwak et al. reported that the decrease in GM3 in the glomeruli of rats after 15 days of STZ administration may cause loss of the filtration barrier and the appearance of glomerular hypertrophy and albuminuria that occur in the early stages of diabetic nephropathy [26]. GM3 is also known to be expressed in podocytes and proximal tubules [13]. Increased GM3 expression has been reported in HK-2 cells, which are a human proximal tubule epithelial cell line, under hyperglycemic conditions [27]. These reports indicate that during hyperglycemia, although GM3 expression in glomeruli is decreased, GM3 expression in the proximal tubules, which comprise the majority of kidney cells, is increased [28]. 

The negative charge associated with sialic acid in GM3 and GD3 expressed in β-cells of the islet may augment the function of glucose transporter isoform-2 (Glut2), which is expressed in the same cells and regulates hyperglycemic load [29]. In this study, enhancement of GM3 via VPA may not affect the function of transporters, such as Glut2, because most of the β-cells in islets were damaged by STZ (Figure 2D). For these reasons, blood glucose levels may not have altered following administration of VPA in this study (Figure 2C). Hyperglycemia represents a key cellular stress in proximal tubule cells by imposing an excessive workload of energy and oxygen requirements, causing progressive kidney dysfunction and fibrosis [30]. However, in this study, we found that no interstitial fibrosis was observed due to hyperglycemia in STZ-induced diabetic nephropathy treated with VPA. From these results, enhanced GM3 expression by VPA elicits a therapeutic effect on not only serum creatinine but also the heightening of interstitial fibrosis (Figure 2F and Figure 3C). Here, we mainly focused on the effect of enhanced GM3 expression on podocytes, but the detailed effect on proximal tubules is currently unknown. Thus, further analysis of the effect of enhanced GM3 expression on proximal tubules may also be warranted in the future.

Using PDGF-β as a mesangial cell marker [22], we observed an expansion of mesangial cells in the glomeruli of diabetic nephropathy mice but an inhibition of this expansion in diabetic nephropathy mice treated with VPA (Figure 5: DN + VPA). An expansion of mesangial cells is associated with the progression of kidney failure in diabetic nephropathy [31]. Note that GM3 is expressed in mesangial cells as well as podocytes of glomeruli in rats [32], but only in podocytes and not in mesangial cells in mouse and human glomeruli [13,17,18]. Therefore, in this study, our results indicated that the observed inhibition of mesangial cell expansion in diabetic nephropathy mice treated with VPA may not be due to the activating effect of VPA on the GM3 synthase gene (*St3gal5*). Moreover, protection of podocytes by enhanced GM3 expression via VPA and the interaction between podocytes and mesangial cells may also inhibit mesangial cell expansion [33,34].

The enhanced GM3 expression via VPA-induced activation of *St3gal5* clearly inhibits albuminuria and glomerulosclerosis in anti-nephrin antibody-induced podocytopathy model mice, and the expression level of GM3 in the glomeruli of MCD and FSGS patients is inversely correlated with the degree of proteinuria [17,18]. In this study, we found that GM3 expression as well as nephrin expression were also inversely correlated with proteinuria in patients with diabetic nephropathy (Figure 7). However, most cases of diabetes mellitus are associated with various factors such as hypertension and obesity in addition to hyperglycemia, resulting in inflammation and oxidative stress [35]. As a result, all cells in the glomerulus (podocytes, mesangial cells, and endothelial cells) are injured, resulting in complicated cell-cell interactions [36,37,38]. Thus, the strength of the correlation between nephrin (or GM3) in podocytes and proteinuria in patients with diabetic nephropathy is easy to reflect the effects of various biases due to the small sample size (Table 1). However, even under these conditions, nephrin (or GM3) correlated with proteinuria, suggesting that GM3 may have a protective effect on podocytes in patients with diabetic nephropathy.

In this study, we have demonstrated that enhancement of GM3 expression via VPA following the onset of diabetes mellitus leads to the inhibition and alleviation of both albuminuria progression as well as the formation of glomerular lesions through the maintenance of podocytes. Namely, the maintenance of GM3 expression via VPA not only suppresses the onset of albuminuria and glomerulosclerosis but also alleviates the progression of diabetic nephropathy, the most common secondary glomerular disease associated with podocyte injury. These findings suggest that maintenance of GM3 expression in podocytes is effective against not only the onset of albuminuria resulting from slit diaphragm injury caused directly by antibody-induced nephrin injury but also indirectly by secondary nephropathy resulting from diabetes mellitus. In addition, there was no observed increase in blood glucose levels in the VPA-only-treated mice (Figure 2C). Hence, the level of enhanced GM3 expression via administration of VPA may be sufficient to delay an increase in albuminuria without inducing insulin resistance [39]. To date, there has been no therapeutic drug that specifically suppresses albuminuria. Our results provide insight into the possible development of such new drugs with few associated side effects for patients with diabetic nephropathy.

## 4. Materials and Methods

### 4.1. Reagents and Antibodies

See Appendix A.

### 4.2. Animal Study

DBA/2JJcl mice (male, 6 weeks old, 19–20 g) were purchased from CLEA Japan Inc. (Tokyo, Japan). Mice were housed in metabolic cages under specific pathogen-free conditions and fed standard chow. For in vivo testing, mice were first randomized into four groups: (i) the control group; (ii) the VPA-treated group; (iii) the STZ-induced diabetic nephropathy group; and (iv) the VPA-treated STZ-induced diabetic nephropathy group (each n = 6, respectively). To induce diabetic nephropathy in mice, DBA/2JJcl mice were given a daily single aliquot of 1.0 mg STZ by peritoneal administration for five consecutive days. VPA was administered as 4 mM VPA in drinking water (100 mg/kg/day as the human equivalent dose). Note that a previous study established that a dosage of 100 mg/kg/day of VPA is safe for humans [18]. Kidney tissues from the four groups of DBA/2JJcl mice were sampled and used for various tissue staining’s. Mice were placed under isoflurane anesthesia for euthanasia.

### 4.3. Biochemical Studies

For diabetic nephropathy therapeutic tests, urine and blood were sampled on each of the following days: 0, 20, 60, and 120 after the final administration of STZ on day 5. Each serum sample was prepared from blood by centrifugation at 1200× *g* for 20 min at 4 °C. Urine albumin, urine creatinine, serum creatinine, and serum albumin were measured using an Albuwell M Test kit (#1011; Ethos Biosciences, Logan Township, NJ, USA), the creatinine Companion (#1012; Ethos Biosciences), and the Creatinine Assay kit QuantiChrom (#DICT-500, BioAssay Systems, Hayward, CA, USA), respectively. Blood glucose levels in the serum samples were measured using a LabAssay Glucose kit (#298-65701, FUJIFILM Wako, Osaka, Japan).

### 4.4. Tissue Staining

Kidney tissues with p57 and periodic acid-Schiff (PAS) staining were assessed as previously described [18,40]. Mesangial expansion of the glomerulus was graded quantitatively by the percentage of glomerular tuft area involvement, as described previously [18]. Platelet-derived growth factor (PDGF) receptor-β staining was carried out using formalin-fixed paraffin-embedded sections of mouse kidney tissues with an optimal antibody. Immunofluorescence staining of nephrin and GM3 was carried out using frozen sections of mouse and human kidney tissues with various antibodies. Pathological images were visualized by optical microscopy (BX51; Olympus, Tokyo, Japan) and analyzed by ImageJ software (ImageJ 1.50i, https://imagej.nih.gov/ij/, accessed on 3 April 2016). Fluorescence images were obtained using a confocal laser microscope (LSM710; Carl Zeiss, Oberkochen, Germany) and analyzed by ZEN imaging software (Zeiss ZEN (blue edition), Carl Zeiss).

### 4.5. Human Study

The methodology used in the clinical study has been described previously [18]. Briefly, normal kidney tissues were obtained from patients who provided written informed consent and were treated by nephrectomy. Kidney biopsy tissues from diabetic nephropathy patients were obtained with opt-out consent. A case-control study was performed using diabetic nephropathy tissues (n = 11) and normal kidney tissues (n = 16), which were obtained from patients who provided written informed consent and were treated by nephrectomy for kidney cancer at Kitasato University Hospital (Department of Urology). Demographic data were obtained from medical records (Table 1). Note that all patients had diabetic triopathy under treatment for diabetes mellitus [41].

### 4.6. Statistical Analysis

The data were analyzed as the mean ± SEM. Statistical comparisons between two groups were evaluated by Mann-Whitney’s U-test. Multiple-group comparisons were evaluated using a one-way ANOVA followed by Tukey’s test. The correlations between albuminuria and GM3/nephrin expression were determined by Spearman’s rank correlation analysis. Statistical analysis was performed by StatFlex Ver.7 (Artech, Osaka, Japan). A *p*-value of less than 0.05 was considered statistically significant.

### 4.7. Study Approval

All animal experiments were approved by the Ethical Review Committee for Animal Experiments of Kitasato University School of Medicine (#2020-016, #2021-010, #2022-032). DNA studies were approved by the Ethical Review Committee for Recombinant DNA Experiments of Kitasato University School of Medicine (#3837). All experiments were performed in accordance with relevant guidelines and regulations, and the animal study is reported in accordance with ARRIVE guidelines. Kidney biopsy tissues (Diabetic Nephropathy) and normal kidney tissues were obtained after approval from Kitasato University Medical Ethics Organization (KMEO) (#KMEO B17-239). This study was conducted in accordance with the Declaration of Helsinki and the ethical principles for clinical studies in Japan.

## Figures and Tables

**Figure 1 ijms-24-11355-f001:**
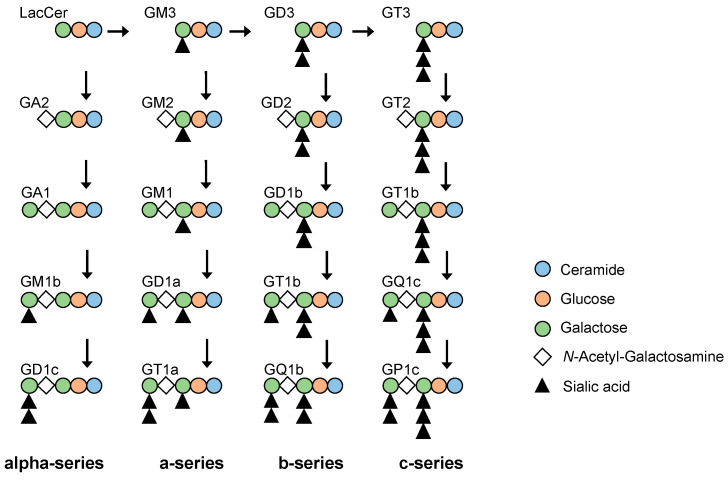
Biosynthetic pathways of gangliosides. Biosynthetic pathway of gangliosides contain alpha-, a-, b- and c-series pathways. GM3 is a derived from lactosylceramide (LacCer) and biosynthesized.

**Figure 2 ijms-24-11355-f002:**
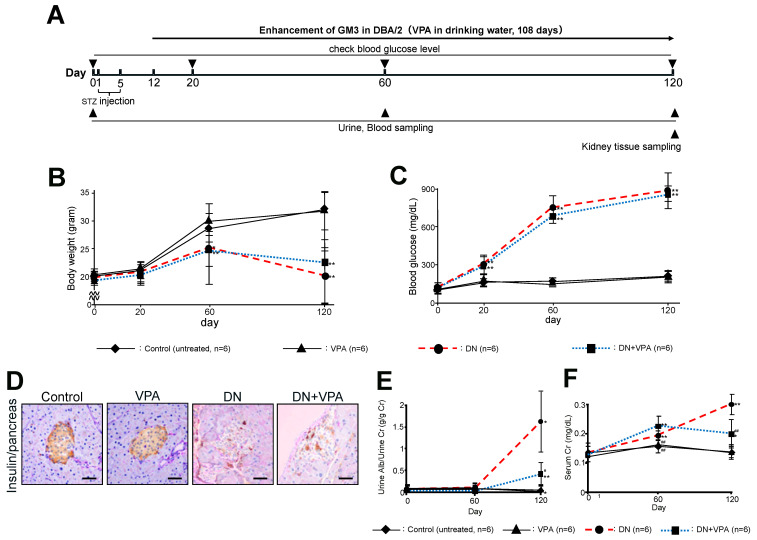
Characterization of streptozotocin (STZ)-induced diabetic nephropathy model mice. (**A**): Schedule for podocytopathy therapeutic test using valproic acid (VPA) in diabetic model mice, (**B**): body weight, (**C**): blood glucose level, (**D**): insulin secretion in pancreatic beta cells. Scale bars: 20 μm, (**E**): line graph showing urine albumin to creatinine, (**F**): serum creatinine. Control: untreated, VPA: VPA-only treated, DN: diabetic nephropathy and DN+VPA: valproic acid treated after occurring diabetes mellitus. ** *p* < 0.01, * *p* < 0.05 vs. Control. ^##^ *p* < 0.01, ^#^ *p* < 0.05; DN vs. DN+VPA. Statistical analyses were performed from mice (n = 6) in each group.

**Figure 3 ijms-24-11355-f003:**
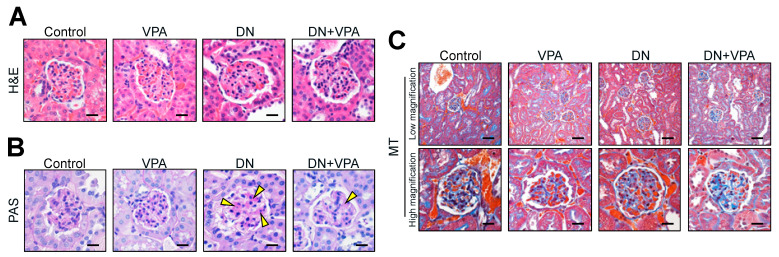
Morphological analysis of kidney glomeruli in various treated STZ-induced diabetic nephropathy model mice. (**A**): Histological analysis by H&E (HE), (**B**): PAS, and (**C**): Masson’s trichrome (MT) staining of kidney glomeruli in 23 week-old mice, respectively. Yellow arrowheads in (**B**) showing glomerulosclerosis. Scale bars: 20 μm (**A**, **B**, and High magnification in **C**), 100 μm (Low magnification in **C**). Control: untreated, VPA: VPA-only treated, DN: diabetic nephropathy and DN+VPA: valproic acid treated after occurring diabetes mellitus.

**Figure 4 ijms-24-11355-f004:**
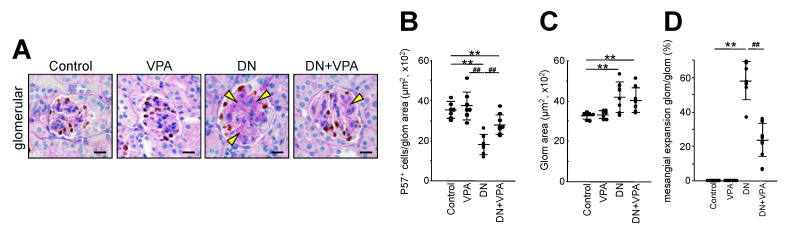
Glomerulosclerosis and inflammation analysis of kidney glomeruli in STZ-induced diabetic nephropathy model mice. (**A**): p57 (podocyte marker, brown) and PAS staining of glomeruli. Yellow arrowheads in (**A**) showing glomerulosclerosis. Scale bars: 20 μm. (**B**): Scatter diagram showing p57^+^ cells/tuft, (**C**): glomerular size, and (**D**): glomerular mesangial expansion/glomerular area, respectively. Control: untreated, VPA: VPA-only treated, DN: diabetic nephropathy and DN+VPA: valproic acid treated after occurring diabetes mellitus. Thirty glomeruli per mouse were analyzed in each dot. Statistical analyses were performed from mice (n = 7) in each group. ** *p* < 0.01 vs. Control ^##^ *p* < 0.01 vs. DN.

**Figure 5 ijms-24-11355-f005:**
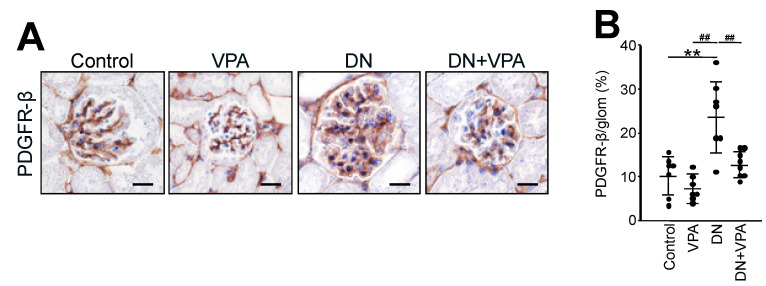
Fibrosis and kidney function analysis of glomeruli in STZ-induced diabetic nephropathy model mice. (**A**) Histological analysis by PDGFR-β (platelet-derived growth factor receptor β) staining of 23-week-old untreated (Control) and variously treated diabetic nephropathy model mice. Scale bars: 20 µm. (**B**) Scatter diagram showing PDGFR-β expression area/glomerular area. Control: untreated; VPA: VPA-only treated; DN: diabetic nephropathy; and DN + VPA: valproic acid treated after occurring diabetes mellitus. Thirty glomeruli per mouse were analyzed in each dot. Statistical analyses were performed from mice (n = 7) in each group. ** *p* < 0.01 vs. Control, ^##^ *p* < 0.01 vs. DN.

**Figure 6 ijms-24-11355-f006:**
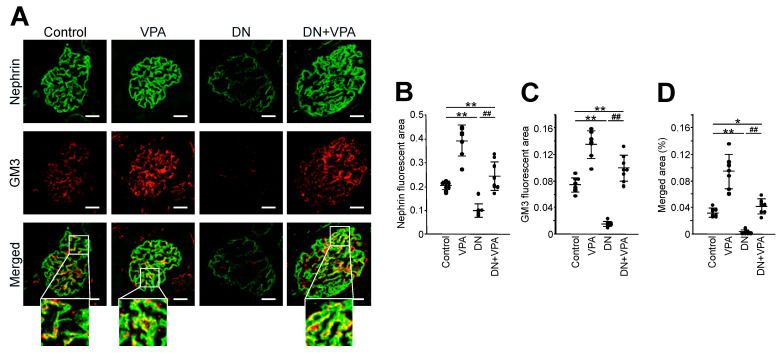
Analysis of GM3 and nephrin expression of kidney glomeruli in STZ-induced diabetic model mice. (**A**): Nephrin (green) and GM3 (red) merged areas (yellow) highlighted in enlarged images. Scale bars: 20 μm. (**B**): Scatter diagram showing nephrin fluorescence area/tuft area, (**C**): nephrin fluorescence area/tuft area, (**D**): GM3 and nephrin merged fluorescence area/tuft area, respectively. Control: untreated, VPA: VPA-only treated, DN: diabetic nephropathy and DN+VPA: valproic acid treated after occurring diabetes mellitus. Thirty glomeruli per mouse were analyzed in each dot. Statistical analyses were performed from mice (n = 7) in each group. ** *p* < 0.01, * *p* < 0.05 vs. Control, ^##^
*p* < 0.01 vs. VPA.

**Figure 7 ijms-24-11355-f007:**
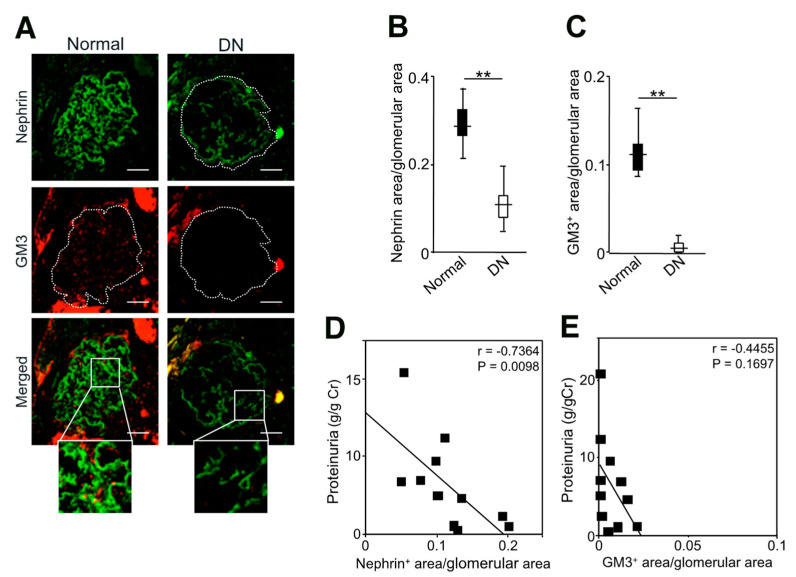
Analysis of GM3 and nephrin expression of kidney glomeruli and correlation between GM3 and proteinuria in patients with diabetic nephropathy. (**A**): Immunofluorescence staining images of nephrin (green) and GM3 (red) merged areas (yellow) highlighted in enlarged images. (**B**): Nephrin fluorescence area/tuft area, (**C**): GM3 and nephrin merged fluorescence area/tuft area, respectively. (**D**): Correlation between nephrin areas (**A**,**B**) and proteinuria, (**E**): correlation between GM3 areas (**A**,**C**) and proteinuria, respectively, Scale bar 40 μm. Normal: healthy subjects, DN: diabetic nephropathy patients. Statistical analyses were performed from healthy subject (n = 16) and patients with diabetic nephropathy (n = 11). ** *p* < 0.01 vs. Normal.

**Table 1 ijms-24-11355-t001:** Baseline characteristics.

	Normal	DN
patients	16	11
male:female	11:5	7:4
age at biopsy, years	62.5 (50, 68.5)	58.0 (50.5, 64.8)
body height, cm	166.9 (158.1, 170.4)	164.2 (159.5, 169.7)
body weight, kg	63.7 (52.5, 69.1)	66.9 (56.3, 79.9)
hemoglobin, g/dL	15.2 (13.4, 15.5)	10.6 (10.1, 12.4) ^##^
serum albumin, g/dL	4.5 (4.3, 4.5)	3.3 (2.5, 3.8) ^##^
serum urea nitrogen, mg/dL	14.6 (12.5, 18.3)	31.4 (23.7, 49.1) ^##^
serum creatinine, mg/dL	0.87 (0.74, 0.93)	2.26 (1.13, 4.21) ^##^
eGFR, mL/min/1.73 m^2^	66.5 (62, 71)	26.0 (10.3, 51.7) ^##^
HbA1c	-	6.0 (5.48, 6.8)
proteinuria, g/gCr	0.04 (0.01, 0.06)	4.94 (1.36, 8.79) ^##^

Data are expressed as median and interquartile range (25th and 75th percentiles). Normal: healthy subjects, DN: Diabetic nephropathy, eGFR: estimated glomerular filtration rate. ^##^: *p* < 0.01 vs. Normal.

## Data Availability

The data that support the findings of this study are available in the main figures and the Appendix A of this article and are also available upon reasonable request.

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
