# Peer review of "Enhanced Levels of Glycosphingolipid GM3 Delay the Progression of Diabetic Nephropathy"

_ijms, 2023, doi:10.3390/ijms241411355_

Round 1

Reviewer 1 Report

Recent studies have highlighted the presence and functions of glycosphingolipids, including GM3, in the kidney. GM3 has been identified in podocyte cell membranes in adult human renal tissue, and it has been shown to be essential for maintaining normal kidney functions through its interaction with nephrin. Podocyte dysfunction is directly linked to the progression of proteinuric chronic kidney disease (CKD), making GM3 a potential therapeutic target for this condition.

This study shows that Administration of VPA decreased levels of albuminuria, glomerular lesions, and inhibited the loss of podocytes and expansion in mesangial area.

Overall, the study highlights the potential of enhanced GM3 expression as a therapeutic approach for diabetic nephropathy by protecting podocytes and inhibiting albuminuria and glomerulosclerosis. Maintaining GM3 expression could have a protective effect against various forms of kidney damage and may offer a new avenue for developing treatments with fewer side effects for patients with diabetic nephropathy. It also shows that VPA has no effect on blood glucose regulation.

1.    Although its known that VPA regulates GM3 synthase expression, it would be ideal to look at GM3 synthase expression in kidney upon VPA treatment.

Reviewer 2 Report

This paper presents very interesting data on the role of GM3 in the progression of diabetic nephropathy (DN) via administration of valproic acid. The effect of valproic acid on kidney function and morphology is impressive, however, the protective role of valproic acid has been described before by several groups. Further, the pathogenic role of GM3 and other gangliosides in the development of diabetic nephropathy has been described.

The paper is difficult to understand, and the authors should concentrate more on the new findings. Nephrin and GM3 in podocytes were decreased in the DN group and its expression correlates with albuminuria in patients with DN. This is an interesting observation, but nothing is known about the underlying mechanism or interaction. The work would greatly benefit from in vitro experiments underlying the hypothesis and the link between valproic acid and GM3. In total, the manuscript should be shortened.

Introduction should be shortened and stream-lined to the experimental settings: clearly discriminate between the effects described and the aim of this study. Number of patients with diabetes are well known and should be removed from the introduction.

Line 85: spelling mistake - enhanced (not “en hanced”)

Line 141: “less marked” for the DN+VPA group – please specify

The authors should reduce to highlight their previous publications, e.g. "Our previous study demonstrated.." should be not more than once or twice used.

Language seems to be ok but the manuscript would benefit from re-phrasing

Author Response

Responses to Reviewer 2 Comments

We would like to thank you for valuable comments, which have helped us to improve the quality of our manuscript. Please consider out point-by-point responses to the Reviewer 2 comments. Please note that in resubmitted manuscript, the words and sentences highlighted in yellow are those that have been revised. Details of the changes are described in the details of the revisions.

Reviewer 2

This paper presents very interesting data on the role of GM3 in the progression of diabetic nephropathy (DN) via administration of valproic acid. The effect of valproic acid on kidney function and morphology is impressive, however, the protective role of valproic acid has been described before by several groups. Further, the pathogenic role of GM3 and other gangliosides in the development of diabetic nephropathy has been described.

The paper is difficult to understand, and the authors should concentrate more on the new findings. Nephrin and GM3 in podocytes were decreased in the DN group and its expression correlates with albuminuria in patients with DN. This is an interesting observation, but nothing is known about the underlying mechanism or interaction. The work would greatly benefit from in vitro experiments underlying the hypothesis and the link between valproic acid and GM3. In total, the manuscript should be shortened.

Introduction should be shortened and stream-lined to the experimental settings: clearly discriminate between the effects described and the aim of this study. Number of patients with diabetes are well known and should be removed from the introduction.

Our response)  Thank you very much for all the advice. As you pointed out, we have shortened the introduction and organized the flow of the text. Also, regarding the description of diabetic patients, we have removed the number of patients and revised it with simpler text. (Revised MS, Line 36, 42, 52, 62, 65, 67-72)

Line 85: spelling mistake - enhanced (not “en hanced”)

Our response) Thank you for your pointing out. We have removed the half-width space inserted in “en hanced”. (Revised MS, Line 73)

Line 141: “less marked” for the DN+VPA group – please specify

Our response) Thank you for your advice. We inserted the statistical data and highlighted that in yellow. (Revised MS, Line 129)

The authors should reduce to highlight their previous publications, e.g. "Our previous study demonstrated.." should be not more than once or twice used.

Our response) Thank you for your comment. We have removed phrases highlighting previous publications and replaced with different phrases. These sentences were also highlighted in yellow. (Revised MS, Line 231)